# The effects of carbon emissions, rainfall, temperature, inflation, population, and unemployment on economic growth in Saudi Arabia: An ARDL investigation

Md Mazharul Islam[1], Majed Alharthi[1], Md Wahid Murad[2]*

**1** Department of Finance, College of Business, King Abdulaziz University, Rabigh, Saudi Arabia, **2** UniSA Education Futures, University of South Australia, Adelaide, South Australia, Australia

☯ These authors contributed equally to this work.

* wahid.murad@unisa.edu.au

**Data Availability Statement:** Data used in the study were collected from the World Bank open data source (https://data.worldbank.org/country/saudi-arabia) and the General Authority of

## Abstract

### Objective

While macroeconomic and environmental events affect the overall economic performance of nations, there has not been much research on the effects of important macroeconomic and environmental variables and how these can influence progress. Saudi Arabia's economy relies heavily on its vast reserves of petroleum, natural gas, iron ore, gold, and copper, but its economic growth trajectory has been uneven since the 1990s. This study examines the effects of carbon emissions, rainfall, temperature, inflation, population, and unemployment on economic growth in Saudi Arabia.

### Methods

Annual time series dataset covering the period 1990–2019 has been extracted from the World Bank and General Authority of Meteorology and Environmental Protection, Saudi Arabia. The Autoregressive Distributed Lag (ARDL) approach to cointegration has served to investigate the long-run relationships among the variables. Several time-series diagnostic tests have been conducted on the long-term ARDL model to check its robustness.

### Results

Saudi Arabia can still achieve higher economic growth without effectively addressing its unemployment problem as both the variables are found to be highly significantly but positively cointegrated in the long-run ARDL model. While the variable of carbon emissions demonstrated a negative effect on the nation's economic growth, the variables of rainfall and temperate were to some extent cointegrated into the nation's economic growth in negative and positive ways, respectively. Like most other nations the short-run effects of inflation and population on economic growth do vary, but their long-term effects on the same are found to be positive.

Meteorology and Environmental Protection, Saudi Arabia open data source (https://www.pme.gov.sa/). The specific searches used by the authors are as follows: GDP growth rates of Saudi Arabia, 1990-2019 Inflation rates of Saudi Arabia, 1990-2019 Unemployment rates of Saudi Arabia, 1990-2019 Population of Saudi Arabia, 1990-2019 Annual rainfall of Saudi Arabia, 1990-2019 Annual average temperature of Saudi Arabia, 1990-2019 Carbon dioxide (CO2) emissions of Saudi Arabia, 1990-2019.

**Funding:** The project was funded by the Deanship of Scientific Research (DSR), King Abdulaziz University, Jeddah, Saudi Arabia under the grant no. G: 112-849-1441. The authors, therefore, greatly acknowledge the funding support from the DSR. The funders had no role in study design, data collection and analysis, decision to publish, or preparation of the manuscript.

**Competing interests:** The authors have declared that no competing interests exist.

## Conclusions

Saudi Arabia can achieve both higher economic growth and lower carbon emissions simultaneously even without effectively addressing the unemployment problem. The nation should utilize modern scientific technologies to annual rainfall losses and to reduce annual temperature in some parts of the country in order to achieve higher economic growth.

## 1. Introduction

Through rigorous research, it is proved that macroeconomic variables have played an important role in economic development that is sustainable in many countries regardless of whether they are still developing or developed [1]. The new millennium has generated more queries and concerns about this issue. As a result, the importance of undertaking more rigorous and in-depth studies is growing. Recently, studies [2–4] have been published on the effects of environmental variables on economic growth, as the impacts of climate change are increasingly noticeable. So, it is reasonable to look at a particular country to produce more specific and robust empirical evidence supporting the argument. These macroeconomic and environmental events can measure the strength of any economy, but less research has examined the outcomes of environmental and macroeconomic variables on economic growth at the country level.

The economy of Saudi Arabia is a factor-driven economy that depends substantially on oil exports which sometimes suffer from external shocks such as fluctuating prices, demand, etc., which has implications for economic growth [5]. Due to the fluctuation of oil prices, Saudi Arabia has experienced erratic growth for the last few decades. The gross domestic product (GDP) percentages were 5%, 10%, 5.4%, 2.7%, 3.7%, 4%, and 1.7% from 2010 to 2016, respectively [6]. This issue has become the focus of discussion among policymakers in recent years as they realized more still needs to be done to sustain future economic growth. To do this, the Saudi Arabian government has a strategy to transform the economy from a factor-driven to an efficiency-driven nation. The Saudi government released its strategic plan in 2016, which included Vision 2030 and the National Transformation Program (NTP) 2020. Focusing on Vision 2030, this statement seeks to diversity the sources of income, economic growth and focus on continuous macroeconomic steadiness so that the economy is no longer so reliant on oil exports. The main goals of Vision 2030 are: increasing the contribution of non-oil products and services (of GDP) from 16% to 50%; enhancing the country's global ranking in the Logistics Performance Index from 49[th] to 25[th] position; and increasing the private sector's contribution from 40% to 65% of total GDP [7]. In last few years, important economic and infrastructure developments and changes have been done in Saudi Arabia such as promoting gender equality, raising the standard of living, improving education, and enacting better health and environmental legislation [7].

The rigorous efforts made by the Saudi government for higher and sustainable economic growth resulted in a moderate acceleration of growth in 2018 by 2.4% compared to 1.7% in 2016 [6]. However, it is believed that not enough progress has been made so far as economic growth declined again to 0.3% in 2019 and then -6.8% in 2020 [8]. This decline could be due to the fall in real output of the oil sector and the recessions in 2019/20 caused by COVID-19. This situation demands changing to a policy of economic diversification in order to make Vision 2030 meaningful. There is potential for gearing growth in the economy, but the main issue is the lack of understanding the effects of environmental and macroeconomic factors. Economic

growth cannot be achieved without understanding what the effects will be. Therefore, it is necessary to discover more about other possible drivers of economic growth to allow for effective and credible policy shifts. In recent years, the effects of environmental and macroeconomic variables on economic growth remain a burning issue among people in Saudi Arabia. Particularly, there is virtually no academic literature (as far as the author knows) that has investigated the long-run relationships among economic growth, inflation, and unemployment, population, rainfall, and carbon emissions in Saudi Arabia. Also, no study has yet looked at the effects of environmental and macroeconomic variables on economic growth of Saudi Arabia, which is economically, legally, politically, culturally, and geographically different from developed and other developing nations. So, the question arises whether changes in any of these factors will contribute to economic growth. The answer to this question in the Saudi context is still unknown.

Thus, addressing this issue is crucial as well as timely for policymakers as they make efforts to diversify the sources of growth and avoid the impact of global shocks. Knowledge about the effects of macroeconomic variables on economic growth will assist policymakers to construct effective legislation that helps achieve Vision 2030. In addition, due to ambiguous relationships among environmental and macroeconomic variables in some nations, this study clarifies the importance of examining empirically the long-run relationships among these variables in Saudi Arabia. Moreover, the lack of a generalized and unified theory on this issue makes it difficult for the Saudi government and its relevant agencies to recognize the lasting effects of carbon emissions, rainfall, temperature, inflation, population, and unemployment on economic growth in Saudi Arabia. This study aims to contribute to the debate by answering following research questions:

1. What are the effects of carbon emissions, rainfall, temperature, inflation, population, and unemployment on economic growth in Saudi Arabia?

2. Are there any causal relationships among these variables?

3. What would be the best policy measure to achieve the goals of Vision 2030?

This study is expected to contribute to achieving the main economic goal of Vision 2030 by finding and explaining the significant (positive and/or negative) determinants of economic growth in Saudi Arabia. This study is also expected to contribute to the literature in the following ways. Firstly, based on our knowledge, this is the first study to examine the long-run relationships between economic growth and environmental factors (e.g. $CO_2$ emissions, rainfalls, and temperature) in Saudi Arabia. Secondly, it contributes to the field (knowledge) in that the data used in the study is up to date, covering the period from 1990 up to 2019.

The remainder of the paper is structured as follows: section two provides a comprehensive literature review followed by section three, which describes the data, analytical techniques, and model specification. The fourth section is devoted to results and discussion, with a focus on the key findings, diagnostic procedures, and strengths of the ARDL model. Section five is the conclusions, summarizing the main themes and implications of the findings. The last section briefly discusses some policy actions based on the key empirical results.

## 2. Literature review

Despite the continuous efforts by researchers to explore the drivers of economic growth there are not yet identified the ultimate determinants that explain why some countries and states grow faster than others. Because economic growth is a complex macroeconomic phenomenon, it is difficult to entirely clarify which environmental and macroeconomic variables

significantly influence economic growth in Saudi Arabia. Generally, economic growth refers to the ability of an economy to accelerate its productivity, for example more capable of producing more goods and services or raising the living standards of its people. Based on the findings of previous studies, it is difficult to distinguish between variables that have greater and important weight in economic growth relationships.

To date, a substantial number of empirical and theoretical studies have investigated the various relationships between environmental and macroeconomic variables. The number is still growing, but no precise consensus on the effects of various environmental and macroeconomic variables on economic growth has yet been reached. Many recent studies, which are reviewed in the following sections, found some thought-provoking effects of environmental and macroeconomic variables on economic growth in various country contexts while other studies found conflicting outcomes. The following section provides a brief review of literature on the dynamics of those environmental and macroeconomic variables and some testable research hypotheses based on the same.

## 2.1 Macroeconomic variables and economic growth

Jilani et al. [9] claimed that inflation has a significant negative effect on GDP growth, which has been supported by several studies [10–19]. Mbulawa [20] reported there is a positive effect of inflation on GDP growth while some other studies concluded that inflation has an insignificant effect on GDP growth [1,21]. Another study conducted by Hasanov [22] discovered a non-linear relationship between economic growth and inflation in the Azerbaijani economy. Similar research had been conducted by Acemoglu et al. [23] on countries that formerly were European powers' colonies; they found secondary negative effects of inflation on economic growth.

Azmi [14] noticed an inverse relationship between economic growth (GDP) and unemployment rate, which has been supported by other studies [12,24–30] while a positive relationship between these factors is reported by Mehmood [31]. Similar research has been conducted by Sadiku et al. [32] on Macedonia where they found no causal relationship between GDP growth and unemployment rate. This finding echoed that of Mucuk et al. [33] who stated that economic growth and unemployment rate are not cointegrated.

Headey and Hodge [34] stated that an extensive number of studies have assessed the relationship between economic growth and population growth. Empirical results of many studies show that the rapid increase in population has a negative effect on economic growth. For example, Linden [35] argues that more people simply reduce long-term potential growth. The findings of Linden [35] have been supported by other studies, including Strulik [36], Prettner and Prskawetz [37], Sabir and Tahir [38], and Bucci [39]. On the other hand, Baker et al. [40] in their US-based analysis concluded that population growth does significantly and positively influence economic growth in that country. However, the results from this study have also been supported by Simon [41], Piketty [42], Sethy and Sahoo [43], Tumwebaze and Ijjo [44], and Garza-Rodriguez et al. [45]. According to World Economics [46], greater population growth leads to more economic growth, scoring more than 3% growth per year for the world. In addition, Zhang [47] stated that unchecked population growth can impede economic development in some developing countries such as China, India, and Indonesia.

## 2.2 Environmental issues and economic growth

Recently, many studies have focused on the determinants of climate change and how it affects economic progress. Consequently, it has become a very interesting topic since environmental conditions play a significant role in politics and economic policy. Studies have examined the

dynamics of overall economic growth and climate change. Recent research concentrating on the effects of the variability of climate on economic growth is varied and diverse [2]. Although the existing empirical literature has provided some evidence on the effects of temperature and rainfall on economic growth, the findings remain inconclusive. For example, Schlenker and Auffhammer [48] stated there is disagreement in the recent empirical literature as to whether temperature affects the level of economic output or economic growth. In addition, Dell et al. [49] have stated that both higher and lower levels of temperature have substantially reduced the level and rate of economic growth regardless of the category of the country, but the poorest countries are severely affected than their richest counterparts. This result has been supported by Barrios et al. [50], Schlenker and Lobell [51], Pindyck [52], Ali [53], Waldinger [54], Odusola and Abidoye [2], Colacito et al. [3], and Sequeira et al. [4]. While the results of the study conducted by Koubi et al. [55] do not produce any evidence that temperature affects economic growth, there is a mixed finding reported by Bernauer et al. [56]. Their study focused on the effects of temperature variability on the growth of an economy. Bansal and Ochoa [57] reveal that temperature does adversely influence overall economic growth both at country and global levels. So, the influence of climate change variability on the growth of an economy remains inconclusive, and the effects differ across tropical areas and climate zones.

Rainfalls are found to have significant economy-wide effects. For example, Berlemann and Wenzel [58] stated that industries are driven by climate conditions such as temperature and rainfall. Carleton and Hsiang [59] reported that environmental conditions, such as temperature, rainfall and violent storms affect economic performance. Some studies have assessed the effects of rainfall fluctuation on sectoral and economic growth by using different analytical models. For example, Sangkhaphan and Shu [60] recently stated that rainfall has reduced growth at the national level through its significant negative effect on agriculture and service sectors, but helped the economies of poor parts of the world, which contradicts the results of Solaun and Cerdá [61]. Ayinde et al. [62] has revealed in their study that an increase in rainfall has a positive effect on economic growth, which has been supported by Miguel et al. [63], Ali [53], Cabral [64], and Odusola and Abidoye [2].

The relationship between environmental degradation and economic growth has emerged as a major theme in modern scholarship, so it is important to examine whether the growing level of $CO_2$ emissions has any effect on economic growth [65]. The extant literature identifies the significant effects of $CO_2$ on economic growth and development, which varies across countries. For example, some studies found a positive and significant impact of $CO_2$ emissions on economic growth in both the short- and long-term [65–68], while other studies reported a negative relationship between $CO_2$ emissions and economic growth [69–71]. Some studies also claimed there is a bidirectional relationship between $CO_2$ emissions and economic growth [72–75], whereas some studies did not find any significant relationship them [76,77].

## 3. Methodology and data description

This section discusses the analytical methods that are employed to address the research questions and achieve the research objective. Brief descriptions and sources of data are also provided in this section.

### 3.1 Description and sources of data

As can be seen from Table 1, the data used in this study are all secondary time series in nature and they are collected from two sources, indicated below in the table. Data for the variables of GDP growth, inflation rate, unemployment rate, population, and $CO_2$ emissions per capita were collected from the World Bank open data source. Meanwhile the data for the remaining

**Table 1. Descriptions of the variables and sources of data.**

| Variables | Elaboration | Data sources |
|---|---|---|
| GDP growth (GDPG) | GDP growth (%) | World Bank [78] |
| Inflation (INF) | Inflation (%) | World Bank [78] |
| Unemployment (UNEMP) | Unemployment (%) | World Bank [78] |
| Population (POPL) | Population in million | World Bank. [78] |
| Rainfall (RF) | Average rainfall in mm | General Authority of Meteorology and Environmental Protection [79] |
| Temperature (TEPT) | Average temperature in Celsius | General Authority of Meteorology and Environmental Protection [79] |
| Carbon dioxide ($CO_2$) | $CO_2$ emissions, tonnes per capita | World Bank [78] |

two variables, including average rainfall and average temperature were collected from the General Authority of Meteorology and Environmental Protection, Kingdom of Saudi Arabia. Data on all the variables are collected for the period 1990–2019.

## 3.2 Unit root test

Table 2 below shows that variables are integrated in a mixed order of I (0) and I (1). This means that the set of variables follow a mixed order of integration using: the following tests for unit root: Augmented Dickey-Fuller (ADF); Phillips and Perron (PP); Kwiatkowski, Phillips, Schmidt and Shin (KPSS). Due to the varied integration of the variables, ARDL bounds testing of cointegration is adopted to determine both the short- and long-run relationships among the variables.

## 3.3 Methodology and model specification

As the main objective of this study is to investigate the effects of Saudi Arabia's environmental and macroeconomic transformations on its economic growth, the essential equation for obtaining the relationships following the Cobb Douglas [80] function is presented below:

$$GDPG = f(CO2, RF, TEPT, INF, POPL, UNEMP) \tag{1}$$

Assuming all other factors as constant the above equation shows that GDP growth (GDPG) of Saudi Arabia is a function of five independent variables, which include carbon dioxide emissions ($CO_2$), rainfall (RF), temperature (TEMT), inflation (INF), population (POPL), and

**Table 2. Unit root test results.**

| Variables | At Level | | | At First Difference | | |
|---|---|---|---|---|---|---|
| | **ADF** | **PP** | **KPSS** | **ADF** | **PP** | **KPSS** |
| GDPG | -4.40*** | -4.40*** | 0.12* | -5.80*** | -7.27*** | 0.13 |
| INF | -2.63* | -2.10** | 0.12 | -8.13*** | -8.02*** | 0.12* |
| UNEMP | -4.22*** | -2.17 | 0.12* | -2.97** | -3.00** | 0.20 |
| POPL | -6.02*** | 2.71 | 0.70** | -4.94*** | -1.28 | 0.45* |
| RF | -0.92 | -1.06 | 0.42* | -4.17*** | -4.19*** | 0.10 |
| TEPT | -3.50** | -3.49** | 0.15** | -6.47*** | -10.10*** | 0.11 |
| $CO_2$ | -4.36** | -2.11 | 0.52** | -4.34*** | -6.90*** | 0.06 |

Notes: ADF = Augmented Dickey-Fuller test for unit root, PP = Phillips and Perron test for unit root, KPSS = Kwiatkowski, Phillips, Schmidt, Shin test for unit root

* indicates significance at the 10% level

** indicates significance at the 5% level

*** indicates significance at the 1% level. Lag Length is based on AIC and Probability is based on Kwiatkowski-Phillips-Schmidt-Shin (1992).

unemployment (UNEMP). Hence, Eq 1 can be transformed into log format here:

$$InGDPG_t = \beta_0 + \beta_1 InCO_2 + \beta_2 InRF + \beta_3 InTEPT + \beta_4 InINF + \beta_5 InPOPL + \beta_6 UNEMP + \mu \tag{2}$$

In the above equation, the log transformation of all the variables, including the dependent and independent, have been done. This is because the symbol '*In*' characterizes the natural logarithm of dependent and independent factors. Furthermore, the coefficients for independent variables are meant to measure by $\beta_1$, $\beta_2$, $\beta_3$, $\beta_4$, $\beta_5$ and $\beta_6$, and the error term will be measured by '$\mu$' in the same equation.

### 3.4 ARDL bounds approaches to cointegration of the variables in the long-run

This study employs the ARDL bounds approach to cointegration in determining the variables' long-run relationships, as the data series have a different order of integration [81,82]. Due to the mixed order of integration of data series (Table 4), ARDL bounds approach to cointegration is deemed to be the most suitable analytical technique for determining the effects of macroeconomic and environmental transformation variables on Saudi Arabia's GDP. The advantage of using this approach to cointegration is that it is very different from ordinary least squares (OLS), vector error correction model (VECM) and vector auto regressive (VAR) techniques due to its short- and long-run estimation provisions. It can integrate self-defined lag length structure. Moreover, the ARDL approach avoids the endogeneity, multicollinearity, heteroscedasticity and autocorrelation problems, which are usually common in other time series regressions. The ARDL model to determine the effects of macroeconomic and environmental transformation variables on GDP in Saudi Arabia are presented in the following equation:

$$y_t = \alpha_0 + \alpha_1 t + \sum_{i=1}^{p} \psi_i y_{t-1} + \sum_{j=1}^{k} \sum_{l_j=1}^{p_j} \beta_j, l_j, x_j, t - l_j + \varepsilon_t \tag{3}$$

Eq 3 portrays the cointegration vectors, where '$y_t$' is predicting the dependent variable, '$\varepsilon_t$' is a scalar zero mean error term, and '$x_j$' is a K-dimensional column vector process, respectively depicted as $\alpha_1, \psi_i$, and $\beta_j, l_j$ in the equation. A constant ($\alpha_0$) is included in the equation but it is neglected here for brevity. Lag operator 'L' is applied to each component of a vector, and polynomial profile is labelled as $\psi$ (L) and (L) $\beta_j$.

## 4. Results and discussion

### 4.1 Descriptive statistics

Descriptive statistics of the whole data set are presented in Table 3 below. It summarizes the values for all the variables, and noticeably, no serious issues are detected with the skewness, kurtosis, and Jarque-Bera normality test results, as they are all within the expected ranges. Some time series data are noticeably skewed to the right or to the left, and the value for kurtosis for some time series data are found to be greater than +1, implying the distribution is too peaked. However, the ARDL bounds approach to cointegration as used in the study, essentially minimizes those extreme outliers, making the measures and inferences reliable and robust.

Additionally, to understand the data set in terms of trends of all the time series variables, two-dimensional graphs of all the seven variables are presented below (Fig 1). Except for the variable $CO_2$, the dependent and five other independent time series variables have demonstrated considerable fluctuation over 1990–2019. However, ultimately, a positive trend of each

**Table 3. Descriptive statistics of the whole data set.**

|  | GDPG | INF | UNEMP | POPL | RF | TEPT | CO$_2$ |
|---|---|---|---|---|---|---|---|
| Mean | 3.54 | 1.88 | 5.80 | 24.32 | 4.53 | 26.39 | 16.19 |
| Median | 2.74 | 1.21 | 5.72 | 23.47 | 4.32 | 26.50 | 16.80 |
| Maximum | 15.19 | 9.87 | 7.34 | 34.26 | 10.59 | 27.74 | 19.52 |
| Minimum | -3.76 | -2.09 | 4.35 | 16.23 | 2.13 | 24.77 | 10.49 |
| Std. Dev. | 4.72 | 2.65 | 0.78 | 5.65 | 2.13 | 0.60 | 2.77 |
| Skewness | 0.89 | 0.94 | 0.34 | 0.30 | 1.27 | -0.26 | -0.63 |
| Kurtosis | 3.49 | 3.89 | 2.75 | 1.79 | 4.20 | 3.59 | 2.26 |
| Jarque-Bera | 4.34 | 5.49 | 0.67 | 2.29 | 9.96 | 0.78 | 2.71 |
| Prob. | 0.11 | 0.06 | 0.71 | 0.31 | 0.01 | 0.67 | 0.25 |
| Obs. | 30 | 30 | 30 | 30 | 30 | 30 | 30 |

*Note: Jarque-Bera test reveals the data normality results; Std. Dev. denotes standard deviation of the variables; Prob. defines the probability of the Jarque-Bera test; and Obs. illustrates the number of observations in the dataset.

variable is noticeable, making the data set suitable for advanced econometric analysis such as ARDL bounds testing of cointegration of the variables under consideration.

In order to select the appropriate ARDL bounds testing model of the long-run equation, it is necessary to determine the optimum lag length (k) by using appropriate model order selection criteria, for example Akaike Information Criterion (AIC), Schwarz Bayesian Criterion (SC) or Hannan-Quinn Criterion (HQ). However, the long-run ARDL model with the smallest AIC and SC estimates or small standard errors and high $R^2$ performs relatively better using standard Vector Auto Regression (VAR). Evidence shown in Table 4 reveals that lag 2 is the optimal lag length (k) of the long-run ARDL bounds model for testing cointegration.

Also, the confirmation for lag selection under the VAR model has been determined in Fig 2, which displays the polynomial graph where all the roots are within the circle confirming that at lag 2, estimates would be appropriate for strong policy decisions and understanding the implications regarding the effects of environmental and macroeconomic changes on Saudi Arabia's economy. Fig 2 also confirms that only two of the fourteen eight roots lie on the unit cycle corresponding to the stochastic trend, while all other roots lie inside. Since no root lies outside the unit circle, the estimated ARDL bounds testing model of long run cointegration appears to be stable.

The ARDL bounds test short-run cointegration estimations, as shown in Table 5, would lead to further decisions of whether there is a short- or long-run cointegration among the variables. Interestingly, whether with a lag order of 1 or 2 and without any lag order all the variables are found to have some varying levels of significant cointegrating relationship. This leads us to the long run cointegrating ARDL model estimation.

**Table 4. Lag order selection criteria: VAR lag order selection criteria.**

| Lag | LogL | LR | FPE | AIC | SC | HQ |
|---|---|---|---|---|---|---|
| 0 | -357.79 | NA | 488.58 | 26.06 | 26.39 | 26.16 |
| 1 | -133.31 | 320.68 | 0.001 | 13.52 | 16.19 | 14.34 |
| 2 | -37.12 | 89.33* | 0.0001* | 10.15* | 15.15* | 11.68* |

Note

* indicates lag order selected by the criterion, LR: Likelihood Ratio test statistic (each test at 5% level), FPE: Final Prediction Error, AIC: Akaike Information Criterion, SC: Schwarz Information Criterion, HQ: Hannan-Quinn Information Criterion.

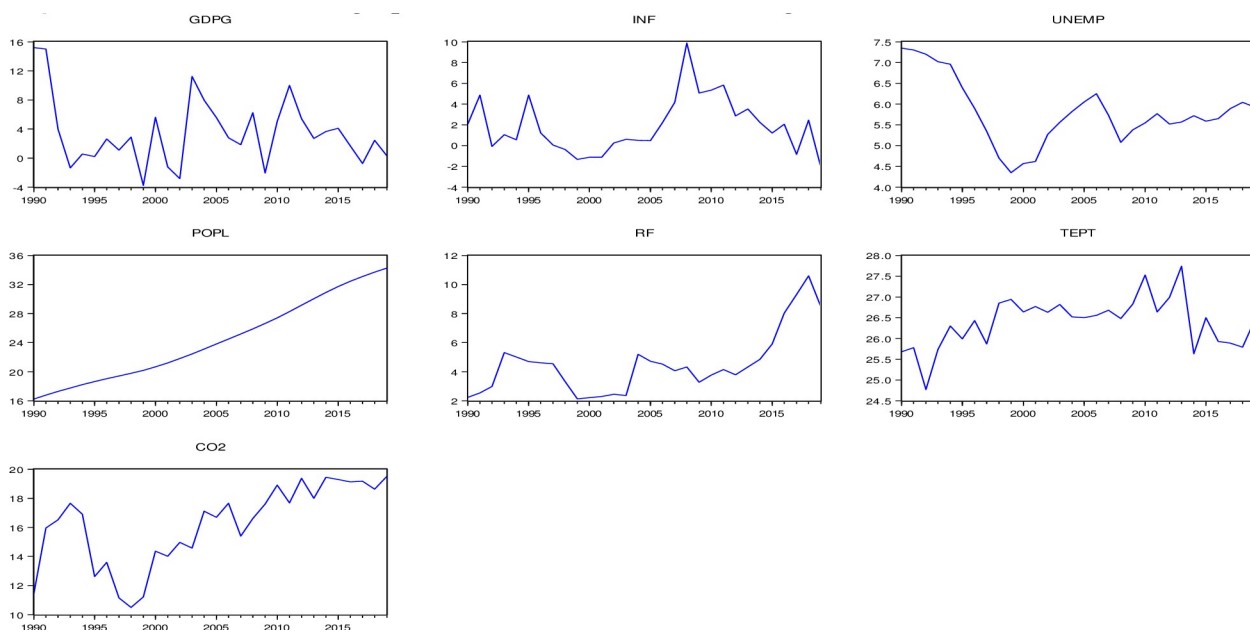

**Fig 1. Two-dimension graphs of all the variables showing the trends of the whole data set.**

Now, to determine the existence of long run cointegrating relationships among the variables under investigation, bound F-statistic (bound test for cointegration) has been computed (Table 6). Under this test, when the computed F-statistic is found to be greater than the upper bound critical value, the $H_0$ is rejected (i.e. the variables are cointegrated). If the computed F-statistic is below the lower bound critical value, the $H_0$ cannot be rejected (i.e. the variables are not cointegrated). As Table 6 shows, the F-statistic value of 12.38 is reasonably large enough to

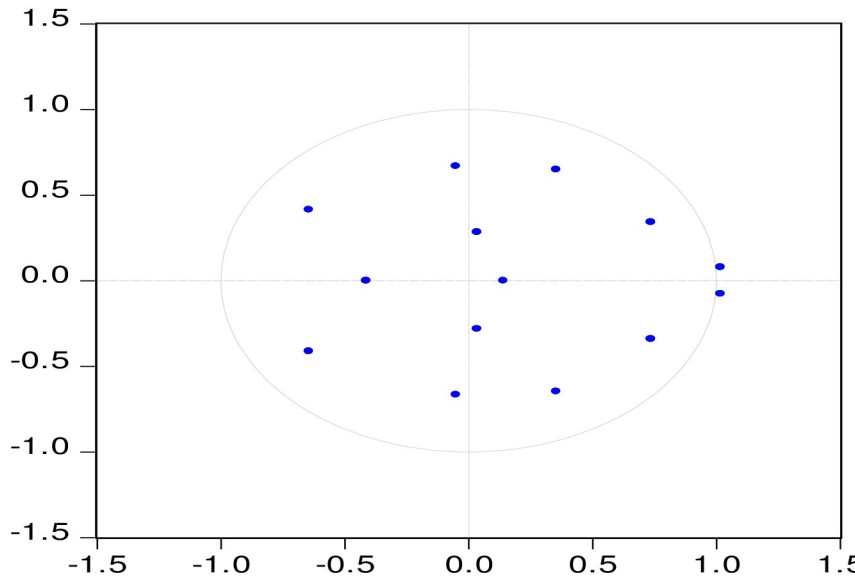

**Fig 2. Lags selection criteria under VAR in polynomial graph.**

**Table 5. ARDL bounds testing results of short run cointegration relationships.**

| Variable | Coefficient | Std. Error | t-Statistic | Probability |
|---|---|---|---|---|
| GDPG(-1) | -0.32* | 0.16 | -1.92 | 0.083 |
| GDPG(-2) | -0.48** | 0.16 | -2.95 | 0.014 |
| INF | 0.89** | 0.33 | 2.68 | 0.023 |
| INF(-1) | -0.70 | 0.40 | -1.73 | 0.113 |
| INF(-2) | 0.30 | 0.33 | 0.90 | 0.385 |
| UNEMP | 7.64** | 3.32 | 2.29 | 0.044 |
| UNEMP(-1) | 9.63*** | 2.77 | 3.47 | 0.006 |
| POPL | 2.58*** | 0.66 | 3.88 | 0.003 |
| RF | -0.52 | 1.04 | -0.49 | 0.628 |
| RF(-1) | 1.12 | 1.00 | 1.11 | 0.292 |
| RF(-2) | -3.28*** | 0.96 | -3.41 | 0.006 |
| TEPT | 3.00*** | 1.33 | 2.25 | 0.047 |
| TEPT(-1) | 2.15 | 1.52 | 1.41 | 0.188 |
| TEPT(-2) | 2.29 | 1.31 | 1.74 | 0.112 |
| $CO_2$ | -0.25 | 0.58 | -0.43 | 0.675 |
| $CO_2$(-1) | -3.32*** | 0.80 | -4.12 | 0.002 |
| $CO_2$(-2) | -1.84*** | 0.52 | -3.51 | 0.005 |
| C | -256.05*** | 78.23 | -3.27 | 0.008 |

*** indicates significance at the 1% level.

** indicates significance at the 5% level.

* indicates significance at the 10% level.

reject the null hypothesis at the 5% significance level. Furthermore, since we have not included a constant or trend in the cointegrating relationship, we can make use of the t-Bounds Test critical values to determine which alternative hypothesis emerges. Hence, the absolute value of the t-statistic is 8.62, which is greater than the absolute value of either the I(0) or I(1) t-bound. Therefore, we reject the t-Bounds test null hypothesis and infer that the cointegrating relationship between the dependent variable and regressors is rational.

Table 7 below portrays the ARDL bounds testing results for the long-term cointegrating relationships among the environmental and macroeconomic variables and how they relate to the Saudi Arabian economy. There are various tendencies detected from the ARDL long-run results. Specifically, five regressors including unemployment, population, rainfall, temperature, and $CO_2$ emissions are impacting on Saudi Arabia's GDP growth significantly. Only one

**Table 6. Bounds testing critical values from Pesaran and Narayan.**

| F-Bounds Test | | | | |
|---|---|---|---|---|
| Test Statistic | Value | Sig. | I(0) | I(1) |
| F-statistic | 12.38 | 10% | 2.12 | 3.23 |
| K | 6 | 5% | 2.45 | 3.61 |
| | | 1% | 3.15 | 4.43 |
| T-Bounds Test | | | | |
| Test Statistic | Value | Sign | I(0) | I(1) |
| t-statistic | -8.62 | 10% | -2.57 | -4.04 |
| | | 5% | -2.86 | -4.38 |
| | | 1% | -3.43 | -4.99 |

**Table 7. ARDL bounds testing results of long run cointegration relationships.**

| Variable | Coefficient | Std. Error | t-Statistic | Probability |
|---|---|---|---|---|
| INF | 0.28 | 0.18 | 1.49 | 0.166 |
| UNEMP | 9.63*** | 1.56 | 6.14 | 0.000 |
| POPL | 1.44*** | 0.34 | 4.17 | 0.001 |
| RF | -1.50** | 0.58 | -2.57 | 0.028 |
| TEPT | 4.15** | 1.71 | 2.42 | 0.035 |
| $CO_2$ | -3.02*** | 0.65 | -4.63 | 0.000 |

*** indicates significance at the 1% level.

** indicates significance at the 5% level.

* indicates significance at the 10% level.

regressor, namely inflation is found to have a positive but insignificant effect on GDP growth. This finding is supported by Mbulawa [20] but it goes against several others, including Martinez and Sancher-Robles [10], Jilani et al. [9], Brixiová et al. [11], D'Costa et al. [15], Azmi [14], Analega and Antwi [12], Kolawole [16], Antwi et al. [13], Ibarra and Trupkin [17], Ramlan and Suhaimi [18], and Mansoor and Bibi [19]. Higher inflation is a likely outcome specifically when the economy is operating near full capacity. On the other hand, a negative relationship between inflation and economic growth may suggest that the economy is operating along the horizontal range of the long-run aggregate supply curve.

Interestingly as well as rationally, only two regressors, namely annual rainfall and $CO_2$ emissions are found to have a significant but negative effect on GDP growth of Saudi Arabia. This implies that both higher annual rainfall and higher per capita $CO_2$ emissions are not conducive for economic growth. There are possible reasons for the negative impact of rainfall and $CO_2$ emissions on the Saudi GDP. For example, higher rainfall could prevent or delay transportation through the effects of flooding, since the infrastructure and general readiness to deal with extensive rainfalls in Saudi Arabia are not yet effective, which leads to less economic activity due to employees finding it difficult to reach their place of work. Another reason for the negative impact of rainfall on economic growth could be that dependence on the agriculture sector in Saudi Arabia is very low compared to its reliance on petroleum resources. According to World Bank [83], the contribution of agriculture, forestry, and fishing to Saudi Arabia's GDP was 2.23% in 2019. The negative impact of rainfall on economic growth is also substantiated by Sangkhaphan and Shu [60] who found rainfall in the case of Thailand negatively impacted on agriculture and service sectors, subsequently reducing economic growth. Higher rainfall is presumably conducive for economic growth, specifically in drought-prone countries. So, the negative effect of rainfall on economic growth, which is not supported by Miguel et al. [63], Ali [53], Ayinde et al. [62], Cabral [64], and Odusola and Abidoye [2], is an extraordinary finding in the context of Saudi Arabia's economy. Regarding the significant but negative long-term impact of per capita $CO_2$ emissions on Saudi Arabia's economy, we find clear and strong support from Ghosh [69], Borhan et al. [70], and Ejuvbekpokpo [71] for India, ASEAN-8 and Nigeria, respectively. While some studies like Saboori and Sulaiman [73], Papiez [72], Zhao and Ren [74], Ghosh et al. [75], and Petrović-Ranđelovića et al. [84] found a bidirectional long-term cointegrating relationship between economic growth and $CO_2$ emissions in the cases of Malaysia, the Visegrad Group countries, China, and Bangladesh, respectively, our finding on Saudi Arabia is clear and robust.

Surprisingly, both unemployment rate and average temperature are found to have a significant but positive effect on the country's GDP. These findings need to be explained with closer

observation of raw data in which it is apparent that since 1998 the unemployment rate and since 1997 the average temperature have both risen in Saudi Arabia while GDP growth also rose during the same period. This implies that the GDP growth of Saudi Arabia is mostly affected by both endogenous and exogenous factors. A similar finding is obtained by Mehmood [31] in the cases of Bangladesh and Pakistan, implying there are countries where an unusual relationship like this exists. A negative impact of economic growth on unemployment rate is not always a likely outcome as demographic factors and institutional conditions in the labor market play moderating roles in the dynamics between the two macroeconomic variables [85]. Eriksson [86] argued that a positive relationship between economic growth and unemployment rate is possible in the long run when economic growth is affected by both endogenous and exogenous factors. For example, excessive reliance on cheaper foreign workers and long-term unemployment compensation and other social support provided to local unemployed workforce by the federal government can moderate the relationship between economic growth and unemployment, and hence a positive relationship is expected. A positive long-term relationship between GDP growth and unemployment rate could also be explained by the proportionate increase in GDP being significantly lower than the proportionate increase in the unemployment rate in Saudi Arabia. Macroeconomic policy effectiveness remains questionable in this regard.

In the case of Saudi Arabia, while the rise in both temperature and GDP growth over the period sounds like they just demonstrate a positive trend, it is considered unrealistic as the relationship differs across tropical areas and climatic zones. This finding is not supported by Bansal and Ochoa [57] who reported a negative relationship. Most studies such as Barrios et al. [50], Schlenker and Lobell [51], Pindyck [52], Ali [53], Waldinger [54], Odusola and Abidoye [2], Colacito et al. [3], and Sequeira et al. [4] found an inconclusive relationship between annual temperature and GDP growth. Also, Dell et al. [49] documented that both higher and lower temperatures significantly reduce the level and rate of economic growth regardless of the category of the country. Also, the world's poorest countries are more severely affected than richer ones. It can therefore be argued that the arid Saudi Arabia does not rely on temperature for its economic growth, but instead on its superior comparative advantage due to enormous stocks of natural resources, including petroleum, natural gas, gold, copper, and iron ore. In no way does the average temperature of the country affect the exploration, processing, production, and exports of these items.

For Saudi Arabia, the nexus between GDP growth and population growth is found to be significantly and positively cointegrated in the long-run ARDL equation. The interpretation is straight forward in that higher population growth is better for long-term economic growth, which coincides with what a number of studies have found, including Simon [41], Baker et al. [40], Piketty [42], Sethy and Sahoo [43], Tumwebaze and Ijjo [44], and Garza-Rodriguez et al. [45]. Only in the cases of some highly populated developing countries, including China, India and Indonesia, did Zhang [47] find that higher population growth undermines economic growth. This could possibly be attributed to those countries' inability to create jobs for the huge number of their unskilled and unemployed labor force. The main consequence of unemployment is the loss of economic output and hence negative GDP growth.

Table 8 summarizes the ARDL bounds testing short-run Error Correction Model (ECM)-based results. The ECM model produced an error correction coefficient for cointegrating equation [(denoted as CointEq (-1)], which was -1.79 and highly significant at the 1% level. However, finding a negative and highly significant ECM-based cointegrating equation coefficient implies a long-term relationship exists between the dependent variable and the regressors. Furthermore, as expected, the error correction term (i.e. model coefficient) represented in the ECM model shown as CointEq(ECM)(-1), is negative with an associated coefficient

**Table 8. ARDL bounds testing short-run ECM based results.**

| Variable | Coefficient | Std. Error | t-Statistic | Probability |
|---|---|---|---|---|
| $\Delta$GDPG(-1) | 0.48*** | 0.09 | 4.91 | 0.000 |
| $\Delta$(INF) | 0.89*** | 0.18 | 4.71 | 0.000 |
| $\Delta$(INF(-1)) | -0.30 | 0.20 | -1.44 | 0.178 |
| $\Delta$(UNEMP) | 7.64*** | 1.46 | 5.20 | 0.000 |
| $\Delta$(RF) | -0.52 | 0.45 | -1.14 | 0.278 |
| $\Delta$(RF(-1)) | 3.28*** | 0.54 | 6.00 | 0.000 |
| $\Delta$(TEPT) | 3.00** | 0.75 | 3.95 | 0.002 |
| $\Delta$(TEPT(-1)) | -2.29*** | 0.72 | -3.17 | 0.010 |
| $\Delta(CO_2)$ | -0.25 | 0.33 | -0.74 | 0.471 |
| $\Delta(CO_2(-1))$ | 1.84*** | 0.35 | 5.24 | 0.000 |
| CointEq(ECM)(-1)* | -1.79 | 0.15 | -11.77 | 0.000 |

*** indicates significance at the 1% level.

** indicates significance at the 5% level.

value of 1.79. This implies two things: firstly, that about 179% of any movements into disequilibrium are corrected for within one period; and secondly, there is a long-run causality running from independent variables to the dependent variable. Most importantly, this finding confirms the dependent and independent variables in the ARDL model are cointegrated and have a long-run relationship.

## 4.2 Model diagnostic tests

An attempt has been made to check the model robustness, and in this process, some diagnostic tests have been conducted to measure the consistency and reliability of the ARDL model and the results. For example, Breusch-Godfrey Serial Correlation LM Test has been conducted to check the model's autocorrelation problem. The test produced Chi-Square statistics of 3.00 with an insignificant probability (Table 9), meaning there is no serial correlation problem in the estimated ARDL model. In the same process, Heteroskedasticity Test of Breusch-Pagan-Godfrey and ARCH test have been conducted and both tests produced insignificant Chi-Square values of 1.11 and 0.25, respectively. This means there is no heteroskedasticity problem in the estimated model. Also, the Ramsey RESET test was done to check the stability of the ARDL model. Since the probability of Chi-Square statistic (0.09) is insignificant the model is considered to be stable. Furthermore, the normality of the estimated model has been checked using the Jarque-Bera test, which produced an insignificant Chi-Square value of 0.33. So, the dataset employed in the study is normally distributed.

The CUSUM test and squared-CUSUM test are used for evaluating the stability of coefficients depicted in Figs 3 and 4, and the graphical picture confirms the outcome of the ARDL

**Table 9. Model diagnostic tests results.**

| Tests | Chi-Square | Probability |
|---|---|---|
| Breusch-Godfrey Serial Correlation LM Test | 3.00 | 0.106 |
| Heteroskedasticity Test: Breusch-Pagan-Godfrey | 1.11 | 0.443 |
| ARCH test | 0.25 | 0.777 |
| Ramsey RESET Test | 0.09 | 0.769 |
| Jarque-Bera test | 0.33 | 0.843 |

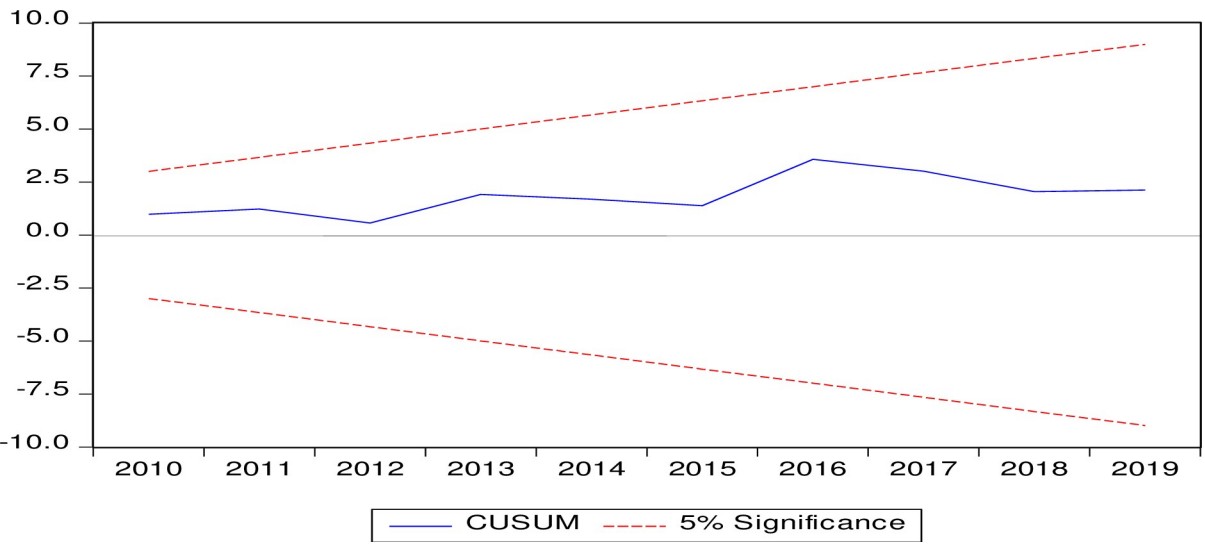

**Fig 3. CUSUM test.**

model. Further down both procedures CUSUM and square CUSUM ensure that the blue line in both figures is inside the red dotted lines. This depiction produces the insight that coefficients are suitable for the dependent variable to predict the future.

To further test the stability of the ARDL model the CUSUM of squares test has also been conducted. It provides a plot of cumulative sum squares of frequency against time and the pair of 5% critical lines. Like the CUSUM test, movement outside the critical lines indicates variable or variance instability. Fig 4 shows that the estimated CUSUM of squares is normally within the 5% level of significance, indicating that the residual variance is fairly stable.

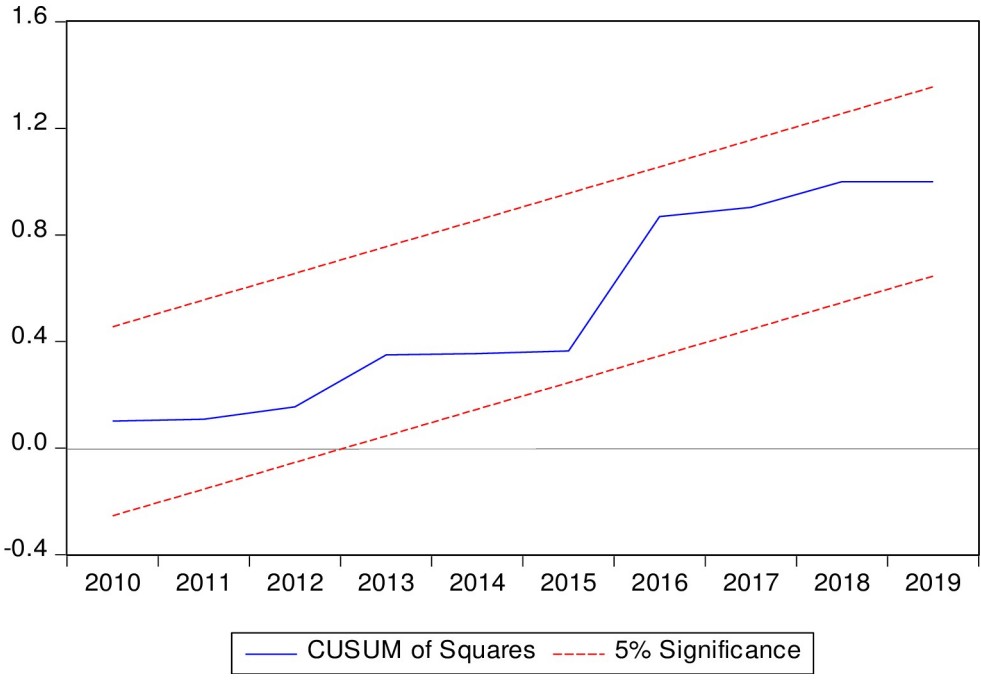

**Fig 4. CUSUM of squares test.**

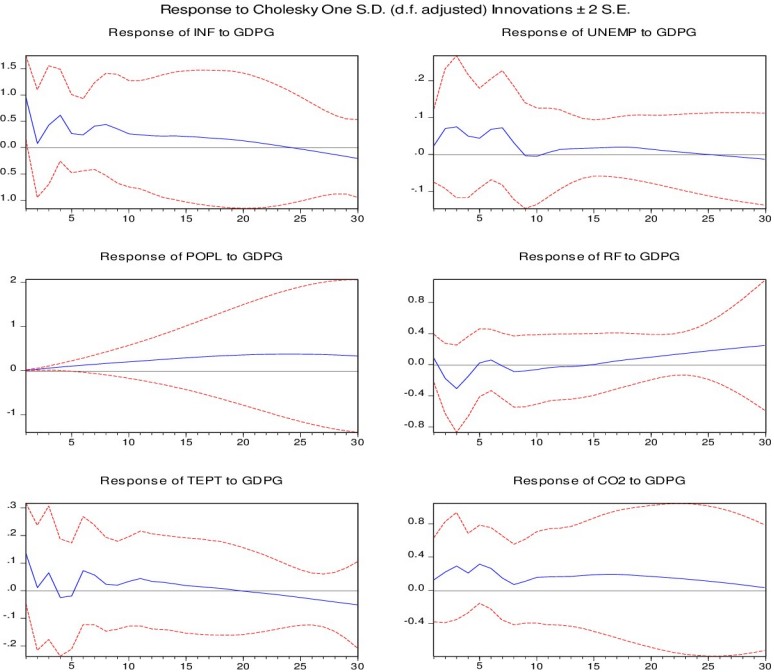

**Fig 5. Impulse response function of variables.**

In this study, the impulse response function (IRF) has been employed to check the exogenous shock of each independent variable to the dependent variable over the study period (Fig 5). Specifically, it is a technique to discover the dynamic effects of all the independent variables individually on economic growth (GDPG) of Saudi Arabia. The IRF of all the independent variables is revealed in a positive one SD shock to the dependent variable. However, as can be seen from Fig 5, an exogenous shock of any independent variable to the dependent variable during the study period has not been detected.

The final diagnostic of the ARDL model variance decomposition of Saudi Arabia's economic growth (GDPG) using Cholesky factors has been estimated, the reason being to check how each random innovation affects the variables. Fig 6 illustrates the results of the variance decomposition of all the variables within a 10-period horizon. The diagnostic test of variance decomposition is suitable for checking how external shocks resonate through an economic system. While economic growth demonstrates a descending trend, other variables specifically carbon emissions, unemployment and inflation have responded to that relatively well. However, the responses of rainfall, temperature and population to economic growth are not decisive.

## 5. Conclusions

This study investigates the effects of environmental and macroeconomic transformations on Saudi Arabia's economic growth for 1990 to 2019. The ARDL bounds testing approach to cointegration, which is very widely used in econometric techniques for time series macroeconomic data analysis, has been employed together with several model diagnostic test to arrive at some precise conclusions. Based on the long-run dynamics of the variables, the most striking but insightful empirical conclusion is that Saudi Arabia can achieve higher economic growth without addressing the nation's unemployment problem. This will continue to be the case as long as Saudi Arabia maintains its huge reliance on a foreign workforce and continues to provide welfare to its own citizens. Its vast reserves of petroleum, natural gas, iron ore, gold, and

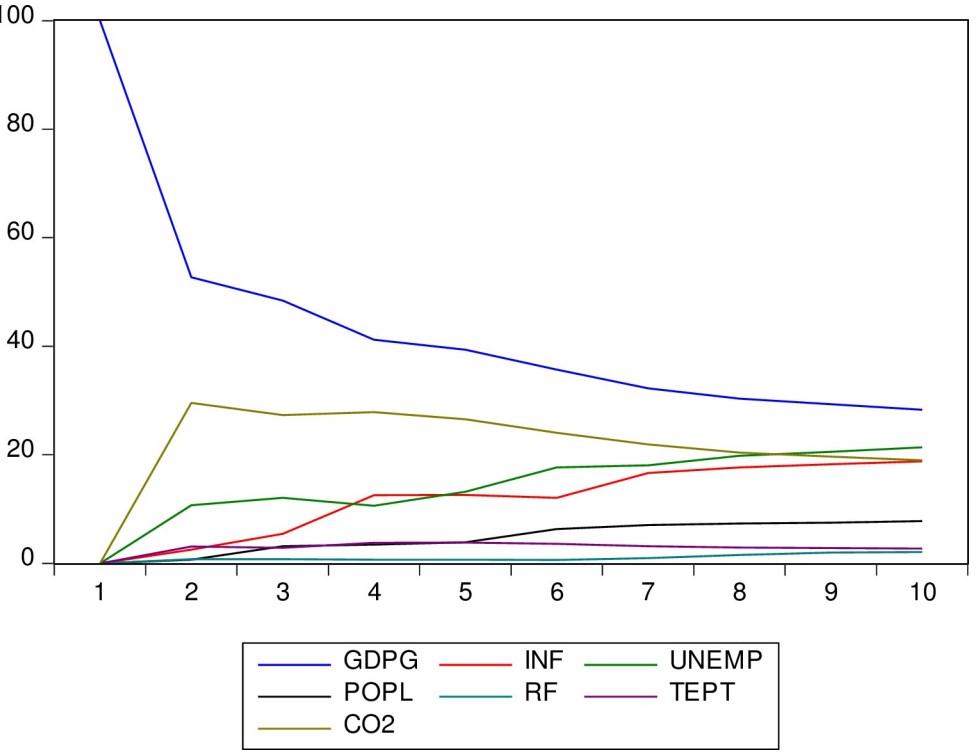

**Fig 6. Variance decomposition of GDPG graph.**

copper coupled are the main contributing factor to the nation's lasting economic growth. While the short-run effects of inflation and population on economic growth are mixed, they both demonstrate a positive effect on the same in the long-term. Interestingly, Saudi Arabia should reduce its carbon emissions considerably to promote economic growth. In other words, moving to a 'green' economic system which involves environmentally friendly renewables would help it achieve sustained economic growth. The effects of the two environmental factors of rainfall and temperature, which are natural as well as largely out of the nation's control, are quite integrated into the nation's economy in the long-term. Therefore, Saudi Arabia should utilize modern scientific technologies to curtail the adverse effects of annual rainfall and temperature in some parts of the country in order to achieve better economic growth into the future.

## 6. Policy actions

Based on the key empirical findings four important policy actions are recommended, and these are expected to positively contribute to long-term economic growth in Saudi Arabia.

### 6.1 Addressing the unemployment problem

The empirical results suggest that achieving economic growth in Saudi Arabia is still possible without addressing the nation's unemployment problem. For decades, abundant petroleum resources have had given the nation a huge comparative advantage, and hence resources-rich

Saudi Arabia can provide enough welfare for unemployed citizens while depending heavily on imported unskilled and semi-skilled foreign workers. Getting citizens into the workforce and getting people to change their expectations must be at the core of economic policy and growth if their lives are to be sustainable. In other words, long-term dependence on the foreign workforce must be reduced. Long-term unemployment of Saudi citizens may lead to widespread social and political disharmony, economic instability, and declining institutional integrity. Ways must be found to address the unemployment of local people if Saudi Arabia is to pursue a market-based economic system.

## 6.2 Retaining comparative advantage with petroleum resources

For Saudi Arabia, having comparative advantage at producing and exporting petroleum resources has been a blessing. Contributions to economic growth by other sectors, including religious-based tourism, foods and accommodation are undeniable but the comparative advantage that is evident with petroleum resources can still help with long-term economic growth. However, this form of comparative advantage will not last forever, and changes in geopolitical, technological, and global circumstances can pose a threat to economic growth in any region. Therefore, Saudi Arabia must find ways to either retain its comparative advantage with petroleum resources or work on building other industries if long-term economic growth is to be viable.

## 6.3 Moving to a green energy system

Fossil fuels are scarce resources and it is universally believed that one day they will run out. For Saudi Arabia, moving from fossil fuels to a green energy-based economic system such as solar energy is a viable alternative for realizing long-term economic growth. This will also help reduce the nation's carbon emissions greatly and promote economic growth in a sustainable way. Saudi Arabia may accept a green energy system now that it is proving to be successful in countries like Denmark, Germany, Norway, and Sweden.

## 6.4 Adopting to variations in rainfall and temperature

Undoubtedly, Saudi Arabia will find it the most challenging task to adapt to variations in rainfall and temperature due to the arid nature of its geography and climate, but the nation must do this to realize economic growth in the long-term. Adverse climatic conditions such as very low or very high rainfall and temperature are not conducive to economic growth. To manage well the adaptation process, however, it should first involve scientists developing research and project partnerships for this aspect of economic progress to occur. While adverse climatic conditions are mostly natural and thus unavoidable, an appropriate and timely adaptation is proven to have minimized any adverse outcomes. Hence, constant monitoring, surveillance, and using modern scientific technologies are vital for achieving positive outcomes.

## Author Contributions

**Conceptualization:** Md Mazharul Islam.

**Data curation:** Md Mazharul Islam.

**Formal analysis:** Md Wahid Murad.

**Funding acquisition:** Md Mazharul Islam, Majed Alharthi, Md Wahid Murad.

**Investigation:** Md Wahid Murad.

**Methodology:** Md Wahid Murad.

**Project administration:** Majed Alharthi.

**Resources:** Majed Alharthi.

**Software:** Md Wahid Murad.

**Supervision:** Majed Alharthi.

**Validation:** Md Wahid Murad.

**Writing – original draft:** Md Mazharul Islam.

**Writing – review & editing:** Md Mazharul Islam, Majed Alharthi.

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
