## [Decision Letter · Decision Letter 0]

4 Jan 2021

PONE-D-20-37757

The effects of carbon emission, rainfall, temperature, inflation, population and unemployment on economic growth in Saudi Arabia: an ARDL investigation

PLOS ONE

Dear Dr. Murad,

Thank you for submitting your manuscript to PLOS ONE. After careful consideration, we feel that it has merit but does not fully meet PLOS ONE’s publication criteria as it currently stands. Therefore, we invite you to submit a revised version of the manuscript that addresses the points raised during the review process.

We look forward to receiving your revised manuscript.

Kind regards,

Ghaffar Ali, PhD

Academic Editor

PLOS ONE

Journal Requirements:

Additional Editor Comments:

Reviewers are not so positive regarding this manuscript they have objections and reservations on almost all sections of this paper. However, I suggest to give authors a chance to improve this paper significantly and make decision after second round. Reviewers have highlighted these points:

1. English language editing is required.

2. Length of the paper is an issue.

3. Introduction section needs revisions in terms of clarity, rationale, problem statement, and objective, etc.

4. Data and methodology section needs plenty of improvement in terms of baseline and analysis of variables.

5. It is also suggested to reduce number of variable because economic growth itself contains almost all economic variables instead of segregating them in the topic and in analyses. Analysis can be revisited comprehensively.

6. Results and discussion can be further improved in the light of comments of the reviewers.

7. Practical policy implications and way forward are missing.

8. Last but not least, please pay serious attention to the author guidelines, currently the paper format is not according to the journal guidelines.

Reviewers' comments:

Reviewer's Responses to Questions

**Comments to the Author**

1. Is the manuscript technically sound, and do the data support the conclusions?

Reviewer #1: Yes

Reviewer #2: No

2. Has the statistical analysis been performed appropriately and rigorously? 

Reviewer #1: Yes

Reviewer #2: No

3. Have the authors made all data underlying the findings in their manuscript fully available?

Reviewer #1: Yes

Reviewer #2: No

4. Is the manuscript presented in an intelligible fashion and written in standard English?

Reviewer #1: Yes

Reviewer #2: No

5. Review Comments to the Author

Reviewer #1: This is an interesting manuscript that covers a range of climate change, social and macroeconomic issues with reference to Saudi Arabia. It discusses an important cluster of numerous questions in Saudi Arabia. I include several comments that should be addressed before publishing this manuscript. Some comments are specific, other are general. The details are given bellow:

1. Length. This paper is very long. In particular, Introduction and review of literature section should be compact and arranged the citation ascending or descending order.

2. The English is better but still requires major editing.

3. Data. Annul Time series data is used from the 1990 to 2019. Data is taken from multiple sources; hence, the base year of data set is not similar. You should mention how you equated the base year.

4. Overall results section is good, but there is need to add some recent studies for cross citation.

5. Add some policy measures that should be based on results

Reviewer #2: In this manuscript authors tried to explore the possible relationships among several climatic-social dimension with economic development.

Overall, the manuscript is poorly written and structured. I am afraid, in its current form, it may not be able to create an interest in readers for global audience.

Abstract, author did not set up a reasonable background of the research objective – the reason why this research is important.

In the introduction, the authors failed to establish a testable hypothesis and interdependencies of the selected variable with the GDP or economic growth.

Methods and Results are poor and do not fit well. There is no discussion and inferences derived from the study.

6. PLOS authors have the option to publish the peer review history of their article (what does this mean?). If published, this will include your full peer review and any attached files.

Reviewer #1: No

Reviewer #2: No

---

## [Author Response · Author response to Decision Letter 0]

13 Feb 2021

We have attached a Word file containing our responses to those comments. Thank you

---

## [Decision Letter · Decision Letter 1]

5 Mar 2021

The effects of carbon emissions, rainfall, temperature, inflation, population and unemployment on economic growth in Saudi Arabia: an ARDL investigation

PONE-D-20-37757R1

Dear Dr. Murad,

We’re pleased to inform you that your manuscript has been judged scientifically suitable for publication and will be formally accepted for publication once it meets all outstanding technical requirements.

Kind regards,

Ghaffar Ali, PhD

Academic Editor

PLOS ONE

Additional Editor Comments (optional):

Reviewers' comments:

Reviewer's Responses to Questions

**Comments to the Author**

1. If the authors have adequately addressed your comments raised in a previous round of review and you feel that this manuscript is now acceptable for publication, you may indicate that here to bypass the “Comments to the Author” section, enter your conflict of interest statement in the “Confidential to Editor” section, and submit your "Accept" recommendation.

Reviewer #1: All comments have been addressed

Reviewer #2: All comments have been addressed

2. Is the manuscript technically sound, and do the data support the conclusions?

Reviewer #1: Yes

Reviewer #2: Yes

3. Has the statistical analysis been performed appropriately and rigorously? 

Reviewer #1: Yes

Reviewer #2: Yes

4. Have the authors made all data underlying the findings in their manuscript fully available?

Reviewer #1: Yes

Reviewer #2: Yes

5. Is the manuscript presented in an intelligible fashion and written in standard English?

Reviewer #1: Yes

Reviewer #2: Yes

6. Review Comments to the Author

Reviewer #1: (No Response)

Reviewer #2: I am OK with the revised version.

The authors have addressed all the comments in the revised version.

7. PLOS authors have the option to publish the peer review history of their article (what does this mean?). If published, this will include your full peer review and any attached files.

Reviewer #1: No

Reviewer #2: No

---

## [Editor Report · Acceptance letter]

24 Mar 2021

PONE-D-20-37757R1 

The effects of carbon emissions, rainfall, temperature, inflation, population, and unemployment on economic growth in Saudi Arabia: an ARDL investigation 

Dear Dr. Murad:

I'm pleased to inform you that your manuscript has been deemed suitable for publication in PLOS ONE. Congratulations! Your manuscript is now with our production department. 

Kind regards, 

on behalf of

Dr. Ghaffar Ali 

Academic Editor

PLOS ONE